# The Effect of the Comprehensive Rural Village Development Program on Farm Income in South Korea

**Eunji Choi [1], Jonghoon Park [2] and Seongwoo Lee [1],*** 

[1]   Department of Agricultural Economics and Rural Development & Research Institute of
     Agriculture and Life Sciences, Seoul National University, Seoul 08826, Korea; ejchoi@snu.ac.kr
[2]   Department of Economics, Hanbat University, Daejeon 34158, Korea; hohojonghoon@hanbat.ac.kr
*   Correspondence: seonglee@snu.ac.kr; Tel.: +82-2-880-4744

**Abstract:** Faced with an aging and declining population, many governments around the world endeavor to revitalize their rural communities in a sustainable manner. In South Korea, the Comprehensive Rural Village Development Program (CRVDP) was carried out from 2004 to 2013 as a key strategy to reinvigorate rural areas. This study aims to conduct an ex-post quantitative evaluation of the effectiveness of the CRVDP in boosting rural households' farm income. In doing so, the present study adopts quasi-experimental research design that is seldom utilized in assessing rural policies. As an alternative evaluation tool with flexibility for using readily available data, the study employed the combined application of the Heckman selection model and the Blinder–Oaxaca decomposition method. The study revealed a significant positive impact of the Program on farm income of rural households in the program-supported areas from both cross-sectional and longitudinal perspectives. A robust causal estimation of the impact of this bottom-up, multi-sectoral rural development program on farm income is achieved, which can be leveraged to widely promote similar type of rural development approach.

**Keywords:** sustainable rural development; policy evaluation; farm income; Heckman selection model; decomposition method

## 1. Introduction

South Korea's (hereafter Korea) rural development policy has evolved to match the changes in the global trend of rural development approaches. The emergence of endogenous growth theories imparted by the seminal works of Romer [1,2] and Lucas [3] in the 1990s marks a notable paradigm shift in rural policy in the modern era, particularly in the advanced economies. The endogenous growth model emphasizes technological change that drives economic growth is generated within the economic system by profit maximizing agents. This theory has been applied in rural policy-making in the midst of a rural decline, following the demise of traditional primary industries since the 1970s. Endogenous rural development emphasizes locally driven, participatory initiatives stimulated by public policy [4–6]. The Organization for Economic Cooperation and Development (OECD) embraced the concept of endogenous rural development in 2006, which marked a paradigm shift in the rural approach, the focus of which has now changed from agriculture to territories and from subsidies to investments [7]. Rural development is no longer seen through an agricultural lens but became a means to enhance competitiveness of rural communities.

Korea also adopted endogenous rural development as a key strategy, beginning in the early 2000s. Concurrent with Korea's fascinating economic growth in a mere half century, the acceleration of

urbanization and industrial development since the mid–1970s has, in turn, led to the marginalization of rural regions. Korea has been experiencing common problems of rural decline, often characterized by the aging and depopulation of young adults. In response to a widening gap not only between urban and rural areas but also between rural areas, the Korean government initiated a rural revitalization policy with an aim to reinvigorate rural areas as vibrant communities. Among a number of different programs that were carried out under the rural revitalization scheme, the Comprehensive Rural Village Development Program (CRVDP) is noted as Korea's representative endogenous and participatory rural development program [8,9]. The CRVDP was implemented from 2004 to 2013 in selected rural villages throughout Korea with goals to improve the residential environment and the income of rural residents. The underlying principles of the CRVDP such as endogenous development, local partnership, and integrated multi sectoral approach are similar to the LEADER+ program of the EU. Since it took a bottom-up approach, contents of the program were different for each participating village, but broader measures included improving social and welfare facilities, promoting rural tourism, and expanding the business bases for income generation and stable production.

The CRVDP was by far the most expensive and extensive rural development program in Korea. However, the importance of policy evaluation on this program has been given minor attention. In fact, compared to government programs in other sectors, such as public health and education, agricultural and rural policies have been relatively free from pressure to apply rigorous evaluation measurements [9]. Since agricultural sectors and rural space are considered as public good providers, the increase in governmental investment was often justified by the multi-functionality of rural areas. In place of scientifically proven, rigorous analysis of program impact, policy evaluation on agricultural and rural programs has employed qualitative approaches or quantitative techniques with some limitations at best. Such observation is also shared in diverse international contexts, especially in Europe [10–12]. Given the massive investment, the lack of rigorous evaluation undermines the credibility of existing assessments and lowers the level of public confidence in policy intervention. In Korea, there is an increasing social demand for improving conditions of rural areas through effective rural policies; at the same time, some critics caution the issue of moral hazard regarding government's massive injection of public funds into rural areas [9]. Therefore, it is imperative to apply rigorous evaluation methods in the assessment of policy effects of rural development programs to resolve the deepening conflict related to resource allocation and to regain the public trust.

To date, the application of rigorous evaluation is scarce in rural policy due to the problem of data, which are often not available at the time of evaluation [10–13]. This study attempts to overcome the problem of data availability when carrying out ex-post impact evaluations of rural development programs by employing alternative evaluation techniques. Thus, the key objective of this research is to provide a robust causal estimation of a rural policy on a targeted outcome using readily available data from ex-post perspective. We selected Korea's representative rural development program, the CRVDP, as a case study and conducted a retrospective evaluation based on a probability model to rigorously assess the effects of the program on boosting the farm income of the rural population. As an alternative flexible quantitative evaluation tool for using a quasi-experimental design, the present study constructs econometrics models incorporating the Heckman selection model and the Blinder–Oaxaca decomposition method. Regression results show that the probability of farm income is higher for farm households whose householder is comparatively older, male, married, with more family members and farming experience, and engaging in cultivation of certain types of crops. However, with the introduction of the CRVDP, age, family size, years of education, and major crop type affected the chance of farm income differently for the program implemented areas and not-implemented areas.

The remainder of the paper is structured as follows: Section 2 introduces Korea's rural development program and ex-post policy evaluation in a rural setting; Section 3 discusses methodologies used for the analysis; Section 4 explains the data and variables utilized in the analysis; Section 5 presents the main results; and Section 6 discusses the results and implications of our findings.

## 2. Research Background

### 2.1. Korea's Rural Development Program in the Period of 2004–2013

In parallel with other OECD member countries, productivist agriculture framed Korea's rural policies until the 1990s. Rural policies were considered synonymous to agricultural policies, which were concentrated on the provision of means to maximize soil yield. It was in the late 1980s and 1990s that rural policies had become increasingly disconnected from agriculture. The top-down or exogenous model of rural development was overtaken by a bottom-up or endogenous philosophy. The premise of the endogenous rural policy is that the development of rural areas should aim at realizing the indigenous potential of endowed natural and human assets [14].

In Korea, the considerations for rural regions as policy targets began to appear on the policy agenda in the late 1990s. In 2004, the government began to seek a goal of promoting quality of life in rural areas as a distinct target from that of agricultural policies with the launch of a "Comprehensive Plan on Agriculture and Rural Communities for 2004–2017" [15]. The budget for the Comprehensive Plan was unprecedented, amounting to 119 trillion Korean Won (equivalent to 108 Billion USD); budgets allocated for two precedent plans in the 1990s were about 40 trillion Korean Won.

Behind the 2004 groundbreaking policy expansion was the exacerbation of socio-economic problems associated with rural decline. Like many other countries around the world, Korea is struggling with the downward spiral of rural decline that arises from typical rural problems such as increasing unemployment, depopulation, economic recession, and a deteriorating quality of life. Aging and depopulation-induced problems, in particular, aggravate the shrinkage of rural communities and local economies. In response, the Korean government promoted the rural revitalization policy as a means to improve the quality of life in rural regions as part of the Comprehensive Plan. As mentioned in the previous section, the CRVDP is a representative program of Korea's rural revitalization policy, which was carried out from 2004 to 2013. The CRVDP was co-financed by the Ministry of Agriculture, Food, and Rural Affairs (MAFRA) and the local autonomy, and MAFRA took charge of supporting 80 percent of the total fund. It was a brand-new rural initiative for the Korean government and a departure from the previous supporting plan, which focused on the expansion of social overhead capital with small-sum, dispersed investments. The CRVDP was designed to sustainably develop rural areas in a way that exploited the diverse geographic, ecological, cultural, and historical characteristics of different rural areas. It concentrated on grouping three to five small villages that share a common cultural identity and similar developmental needs under one hub village. When selected, a hub village could receive 4 to 7 billion Korean Won (3.6 to 6.4 million USD) for up to four years.

The CRVDP has four pillars: (1) Expanding bases for income generation; (2) improving rural landscape; (3) improving residential facilities; and (4) building business management capacity. Each pillar has an array of policy instruments, and all these measures were expected to directly or indirectly contribute to boosting income for supporting areas. For instance, the support for income generation include construction of energy facilities for eco-friendly agriculture, agro-processing and storage facilities for organic products and local specialties, development of farmers' markets, and stimulation of rural tourism. The selection and implementation process of the CRVDP consists of several stages. First, residents prepare and submit a proposal to the provincial government for deliberation through the local autonomy. Second, program implementation villages are selected by the MAFRA after the final deliberation. Third, a general plan is finalized through public hearings and in consultation with local public entities. Fourth, an implementation plan is approved by the provincial government and the MAFRA. When a proposal is received, the local autonomy prepares the investment portfolio, which should be allocated to reflect the goals of all four pillars [16]. Moreover, the deliberation process involves site visits, program information sessions, and evaluation by a group of 25 experts.

Given the large volume of investment, however, the agricultural and rural policies to renew rural communities in Korea are confronted with harsh criticisms. Six years have been passed since the

termination of the CRVDP in 2013. However, questions on the efficiency of the CRVDP are still being raised when it comes to the issue of the effectiveness of rural policies. One of the concerns being raised with respect to the CRVDP is that the program had been heavily concentrated on building hardware facilities in lieu of expanding welfare services or leisure programs for rural residents. In the absence of rigorous evaluation on the effects in measurable terms, the effectiveness of this government policy will remain controversial, which makes it difficult to design future programs with appropriate policy targets. Korea's agricultural and rural policies have been relatively free from rigorous evaluation measurements, largely because of the sensitivity in dealing with rural areas that are commonly seen as nostalgic hometowns where traditional values are preserved. As a result, providing support to rural areas has been taken as a societal obligation in which positive externalities are being produced for the benefit of the whole nation. While government actions to deal with the rural decline is justified, conducting scientific and objective evaluations on government interventions remains imperative.

### 2.2. Ex-Post Quantitative Policy Evaluation in Rural Development

Drawing a causal inference of a policy intervention on the intended targets is an intricate task since various immeasurable factors that are not directly related to a particular policy also affect its outcomes. The assessment of rural development programs is even more challenging, given the complex and cross-sectoral nature of rural communities [10].

A robust evaluation requires the exploitation of counterfactual situations through which an objective comparison of outcomes between the treatment and control groups can be made [17]. Unlike most commonly applied ex-post evaluation techniques in the agricultural and rural sector, such as survey-based outcome monitoring or cost-benefit analysis, counterfactual methodologies explore the causal impact of a policy on the outcomes of interests. Policy evaluation results based on the opinions of the beneficiaries or expert judgement are pointed out as "guesstimates" [12]. On the other hand, a policy evaluation entailing proper comparisons enables valid estimates of the causal effect fully attributable to the policy intervention [10,17,18].

The most valid counterfactual analysis is possible by setting up an experiment through the randomized controlled trial (RCT) or by finding a natural experiment. While the latter may involve a tremendous effort as well as some luck to find dichotomized groups that exist by nature, the experimental-based RCTs, often considered as the golden standard in evaluation, are enormously costly in terms of money and time. The two approaches are limited in practicability in actual application and cannot be employed retrospectively. Ex-post evaluation of an intervention exploiting the counterfactual is possible through quasi-experimental methods using observational data [19–21]. The quasi-experimental approach is designed to estimate causal impact of an intervention in a way that resembles the experimental research without the involvement of random assignment. The most commonly applied quasi-experimental identification strategies include difference-in-differences (DID), regression discontinuity (RD), instrumental variable (IV), and matching [22,23]. Quantitative estimates attained by converting intangible observations into tangible effects have a particular merit in persuading the public about the program's effects. A number of prominent studies particularly in education, public health, and labor economics used these methodologies to estimate causal effects of public policies on outcomes. For example, the DID methodology has been utilized to study the impact of a rise in the minimum wage on employment [24]. The IV approach has been applied to evaluate the impact of charter schools on student achievement [25]. An RD design was used to examine the effect of alcohol consumption on mortality using the minimum drinking age as a group assignment variable [26]. Among matching methods, the PSM approach is the most developed and popular strategy to create a control comparator population [27] while the synthetic control approach is gaining popularity in recent years [28,29]. Despite the usefulness, however, the application of such methodologies is seldom in practice.

There are two major contributing factors behind the limited application. One is a policy planning culture within the government that places top priority on policy making, which makes the ex-post

policy evaluation the "forgotten phase of planning" [12]. Policy makers tend to place a higher priority on policy planning to strategically win the popular support and establish political legitimacy. Another factor is the data availability at the moment of evaluation, which mostly depends on past policy decisions [10,12,13]. The most commonly adopted quasi-experimental methodologies require an extensive coverage of databases. For instance, the parallel trend assumption of DID requires the source of selection bias to be time-invariant whereas the omission of any crucial variable will lead to biased results in the case of the propensity score matching [30,31]. Also, finding a valid instrumental variable is quite difficult in practice [32,33]. Thus, the application barriers of the existing quasi-experimental evaluation methodologies highlight the paucity of credible methodologies; this prompts the need to search for alternative methodologies that can overcome such hurdles.

The same problems are also evident in ex-post evaluation of rural policies [10,34,35]. There has been a rising call to come up with evidence-based recommendations employing scientifically credible evaluation methodologies in the agricultural and rural sector [35–38]. However, rural policy evaluators have been embroiled in the problem of the lack of relevant data to apply such methodologies. Seeking credible combinations using a wide variety of existing quantitative methodologies can be one valid approach in the search for an improvement in evaluation tools. In this context, this research is one attempt to fill such methodological lacuna to ultimately support evidence-based policy-making.

## 3. Methodology

The present study conducts an ex-post quantitative evaluation of the CRVDP based on the assumptions of counterfactual reasoning. In the case of a policy intervention in which the treated group and the untreated group were not established at the initial stage such as the CRVDP, it is not possible to construct a control group in the ex-post setting. In this case, researchers are tasked to establish a valid comparison group that is theoretically and statistically reasonable. This study attempts to empirically analyze the effect of the CRVDP on farm income by setting up a counterfactual comparison group in a way that, we can reasonably assume, is equivalent to the sample who did not benefit from the policy implementation.

As a novel way of evaluating the impact of rural policy on the farm income of rural households by using secondary data such as a census, this study employs the counterfactual decomposition technique introduced by Blinder [39] and Oaxaca [40]. The Blinder–Oaxaca decomposition allows quantification of the treatment effect by dividing the outcome differentials between the two groups into an "explained" part due to differences in observed characteristics and an "unexplained" part attributable to net effects of group membership [41]. However, the conventional decomposition technique does not address sample selection, which could result in over or underestimation of true effects. In this study, selectivity bias is corrected by employing the Heckman selection model applied previously to the decomposition analysis.

### 3.1. The Heckman Selection Model

The data constructed for this study divide Korea's entire farm households into either a program implemented group or not-implemented group, according to whether or not the residential locations of farm households belong to one of the program implemented areas. The CRVDP is a bottom-up program in which individual villages centered around a hub village apply to receive support from the government. Therefore, self-selectivity is inherent because participation depends on the choice of individual farm households. Certain villages composed of particular groups of households may have decided to take part because of the household's self-selecting traits. For instance, if the initial inferior conditions of participating areas had affected their chance to be selected by the government, then the outcomes found in the selected areas could be influenced by their low potential. In this case, evaluating program impacts by comparing the treated areas with the untreated areas is prone to sample selection bias with respect to observable and unobservable variables [42–44].

Moreover, it is unrealistic to assume that the same farm households are maintained over the analyzed period, especially in the rural development context. For instance, rural demographics could have been affected by a popular trend of urban-to-rural migration. Or, the introduction of new IT devices such as smartphones may have affected the composition of the sample. If farm households drop out or newly join in a biased way, then the farm households in the program implemented areas in the post-program period are unlikely to be representative of the original sample pool in the pre-program period.

The Heckman selection model is a statistical method that effectively resolves potential biases caused by the selection process [42,45]. The intuition behind this methodology is that sample selection bias arises because the probability of selection into the sample is omitted from the regression model. Therefore, Heckman [45] addresses this problem by treating the selection problem as a special case of the omitted variable problem and corrects the bias by simply adding a regressor to the regression model that reflects the probability of selection into the sample. In order to do so, the first step is to run a probit regression model to compute the inverse Mill's ratio (IMR), $\lambda_i = \frac{\phi(W_i\alpha_i)}{\Phi(W_i\alpha_i)}$, as a correction factor [42]. Then, as a second step, the IMR is included as an additional control variable in the outcome as in Equation (1):

$$\mathrm{E}[y_i|x_i,\ in\ sample] = \beta'x_i + (\rho\sigma_\varepsilon)\lambda_i = \beta'x_i + \theta\lambda_i \tag{1}$$

where $y_i$ denotes the outcome variables, $x_i$ denotes the observable features of the independent variables, $\beta$ denotes the parameters to be estimated, $\rho$ is the correction coefficient, and $\sigma$ is the variance of the error terms (see Appendix A for a detailed discussion of the methodology).

In this study, a multiple regression is carried out using the estimated coefficients of the maximum likelihood estimation (MLE) model. The MLE model is more widely used than the simple ordinary linear (OLS) model since more efficient estimates can be attained [46,47]. In the Heckman selection model using the MLE, the correction factor $(\lambda)$ is expressed as the product of rho $(\rho)$ and sigma $(\sigma)$.

### 3.2. The Blinder–Oaxaca Decomposition Technique

Once potential selection bias is addressed, the net impact of a policy can be estimated under the Blinder–Oaxaca decomposition framework. This method has been most frequently applied in labor economics to explain wage differentials between naturally discrete groups such as males and females, immigrants and natives, and black and white workers. The Blinder–Oaxaca decomposition identifies underlying causes of inter-group differences by breaking down those differences into observable characteristics or "endowments," and those not clarified by the endowed characteristics or "residuals." In this light, the Blinder–Oaxaca decomposition can be applied in policy evaluation to estimate the net policy impact [34,48]. The decomposition technique allows the identification of how much of mean differences on outcomes across two groups can be explained by the differences in observed characteristics. The rest of differences that cannot be explained by observed characteristics can be defined as exogenous effects. When assumptions of selection on observables or unconfoundedness are satisfied [27,49], counterfactual effects from decomposition can be interpreted as causal effects [41,50]. Therefore, in policy evaluation, exogenous effects from the decomposition analysis can be interpreted as the average treatment effect of a policy.

The Blinder–Oaxaca decomposition is a convenient way to quantify the separate contributions of group differences in terms of observed characteristics [51]. The particular advantage of using the decomposition technique in place of the most widely applied quasi-experimental methods mentioned previously is that policy outcomes can be identified either from cross-sectional or longitudinal perspectives. A cross-sectional analysis compares the outcomes of a policy intervention between a treatment group and a comparison group, while a longitudinal analysis measures a change in outcomes of pre- and post-intervention. In this study, the outcomes are compared from both cross-sectional and longitudinal perspectives to ensure the validity of the analysis.

To identify the net policy impact using the Blinder–Oaxaca decomposition, the linear regression defined by Equation (1) is divided into the treatment group ($E(Y_A)$), and comparison group ($E(Y_B)$), as shown below:

$$Group\ (A): E(Y_A) = \sum_{j=1}^{k} \beta_j^A \overline{X}_j^A \tag{2}$$

$$Group\ (B): E(Y_B) = \sum_{j=1}^{k} \beta_j^B \overline{X}_j^B \tag{3}$$

where $\beta$ is the vector of coefficients and $\overline{X}$ is the mean of independent variables.

In a cross-sectional analysis, Equation (2) applies to the area (A) where the program had been implemented (treatment group), and Equation (3) applies to the area (B) where the program had not been implemented (comparison group). In this case, a comparison of outcomes is made between farm households in the program implemented areas and farm households in the program not-implemented areas. On the other hand, in a longitudinal analysis, changes in outcomes before and after implementation of the program on farm households in the program implemented areas are compared. Thus, Equation (2) is for the group in (A) period after the program had been implemented, whereas Equation (3) is for the identical group in (B) period before the introduction of the program. Since the Equations (2) and (3) are defined as forms of the expected value, the expected differences between the two groups can be directly compared in both cases. The mathematical expression of this theoretical concept is as follows:

$$E(Y_A) - E(Y_B) = \sum_{j=1}^{k} \beta_j^A \left( \overline{X}_j^A - \overline{X}_j^B \right) + \sum_{j=1}^{k} \overline{X}_j^B \left( \beta_j^A - \beta_j^B \right) \tag{4}$$

The first part of the Equation (4) shows the total effect expressed in terms of the endowment effect and the residual effect (see Appendix B for mathematical details). In a cross-sectional analysis, the endowment effect is the effect produced by the initial differences in observed characteristics across two groups in the same time period. In contrast, in a longitudinal analysis, the endowment effect is the differences in observed characteristics across the pre- and post-implementation samples in the beneficiary areas that are unrelated to the program implementation. Therefore, the endowment effect is the effect not directly related to the program implementation and is denoted in the first term on the right-hand side of Equation (4). On the other hand, the second part of this Equation is the residual effect that indicates the difference in the outcomes between the two groups generated by the program implementation. The residual effect is the effect reflected by factors other than the differences in the independent variables between the two groups in comparison; thus, the residual effect captures the average treatment effect.

## 4. Data and Variables

The data for the analysis were collected from the Korea Agricultural Census from two different years: 2005, before the implementation of the CRVDP, and 2015, after the completion of the program. This data provided by Statistics Korea contain a set of micro-level individual and household characteristics of all Korean farm households.

In Korea, there are a total of 1388 rural villages of *eup* and *myeon* administrative districts. In order to discern districts where the CRVDP had been carried out, we acquired internal data of the MAFRA with a full list of administrative districts who participated in the CRVDP through the Rural Development Corporation, a major program implementation agency. From these data, we were able to identify 301 local villages at eup/myeon administrative level where the program had been carried out during the period of 2004–2013. These data were matched with the 2005 and 2015 agricultural censuses,

through which we were able to dichotomize farm households into the treatment group and the comparison group.

The original census data had 1,272,882 farm households in 2005 and 1,087,843 farm households in 2015. In order to obtain balanced sample sizes between two study regions, we applied a 10% sampling for the program implemented areas and a 5% sampling for the program not-implemented areas on a random basis. As a result, the final sample drawn from the 2005 census contains 21,951 farm households in the policy implemented areas and 47,062 farm households in the policy not-implemented areas. The final sample obtained from the 2015 census includes 17,686 farm households in the program implemented areas and 39,344 farm households in the program not-implemented areas. Moreover, in this study, the target of the analysis was limited to farm households whose householders were 19 years old or older at respective time periods. The descriptive statistics of the final samples are presented in Appendix C.

Table 1 displays a description of dependent and independent variables of this study. Reflecting the structure of the Heckman selection model, the first dependent variable in the binomial probit model is the probability of selection into the treated sample. In the cross-sectional analysis, it is the probability of an individual household to belong to the program implemented group. On the other hand, in the longitudinal analysis, it can be expressed as the probability of being in the sample of post-program period. In the second stage of the Heckman selection model, the dependent variable is the farm income described by the sales amount of agricultural and livestock products (the Consumer Price Index is used to adjust farm income differences between 2005 (=78.4) and 2015 (=100).). Since the sales amount of agricultural and livestock products were coded in categorical format, the data were linearized by the median value of the sales taking a natural logarithm.

**Table 1.** Description of variables.

| Variable | | | Description |
|---|---|---|---|
| Dependent Variables | | | |
| | | | (Cross-sectional) First stage = Policy implemented area (=1), Policy not-implemented area (=0) (Longitudinal) First stage = After policy implementation (=1), Before policy implementation (=0) |
| | | | Second stage = Log transformation of farm income |
| Independent Variables | | | |
| Demographic | Age of householder | AGE1 | 19~34 (=1), otherwise (=0) (Ref.) |
| | | AGE2 | 35~44 (=1), otherwise (=0) |
| | | AGE3 | 45~54 (=1), otherwise (=0) |
| | | AGE4 | 55~64 (=1), otherwise (=0) |
| | | AGE5 | Over 65 (=1), otherwise (=0) |
| | | AGE | Householder's age (linear) |
| | | AGE_SQ | AGE*AGE |
| | Gender | MALE | Male (=1), Female (=0.) |
| | Marital status | MARRY | Married (=1), otherwise (=0) |
| | Number of family members | HHNUM1 | 1~2 (=1), otherwise (=0) |
| | | HHNUM2 | 3~4 (=1), otherwise (=0) |
| | | HHNUM3 | Over 5 (=1), otherwise (=0) (Ref.) |
| | | HHNUM | Household size (linear) |
| Socio-economic | Education | EDU1 | Below high school (=1), otherwise (=0) |
| | | EDU2 | High school diploma or some college (=1), otherwise (=0) |
| | | EDU3 | BA or higher degree (=1), otherwise (=0) (Ref.) |
| | | EDUY | Years of education (linear) |
| | | EDUY_SQ | EDUY*EDUY |
| | Experience in farming | EXP1 | Under 10 years (=1), otherwise (=0) (Ref.) |
| | | EXP2 | 10~20 years (=1), otherwise (=0) |
| | | EXP3 | Over 20 years (=1), otherwise (=0) |
| | | NEW | Less than 6 years (=1), otherwise (=0) |

**Table 1.** *Cont.*

| Variable | | | Description |
|---|---|---|---|
| Agriculture | Property | Machine | Possession of agricultural machinery (=1), otherwise (=0) |
| | | Info | Computer usage (=1), otherwise (=0) |
| | | Other | Participation in other agriculture-related businesses (=1), otherwise (=0) |
| | Crop | CROP1 | Rice (=1), otherwise (=0) |
| | | CROP2 | Fruit (=1), otherwise (=0) |
| | | CROP3 | Other type of crops (=1), otherwise (=0) |
| | | CROP4 | Upland crop (=1), otherwise (=0) (Ref.) |
| | | CROP5 | Livestock (=1), otherwise (=0) |
| | Sales Place | S_PlACE1 | Wholesale market, production market (=1), otherwise (=0) |
| | | S_PlACE2 | NH (Korean agricultural cooperative), agricultural corporation (=1), otherwise (=0) |
| | | S_PlACE3 | Government, collector, mediator (=1), otherwise (=0) |
| | | S_PlACE4 | Direct sales to consumers (=1), otherwise (=0) (Ref.) |
| | | S_PlACE5 | Retailer, agricultural processing company (=1), otherwise (=0) |

The independent variables were classified into demographic, socioeconomic, and agricultural management characteristics of farm households. We selected the probable determinants that affect the likelihood of earning farm income based on the previous literature and information available in the census. Demographic factors such as age, gender, marital status, and household size are the most important factors that determine farm income [9,52]. The positive effect of agricultural experience on farm income is quite obvious while education is often proved to be positively associated with farm income [53]. In terms of agriculture-related properties, it is highly likely that the possession of agricultural machinery and utilization of computer would positively affect farm income since such factors enhance agricultural productivity and rate of returns [54]. Participation in agriculture-related business is expected to positively influence farm income through the formation of agricultural value chain [55]. Type of agricultural cultivation activities and marketing channels in general determine the level of agricultural profit, and thus, these factors should be closely related to farm income [34,52,56].

## 5. Empirical Results

### 5.1. Average Farm Household Income by Policy Implementation

Prior to analyzing the effect of the CRVDP on farm income using the sample population drawn randomly from the census data, we compared the simple difference in mean earnings between the program implemented areas and not-implemented areas using the original censuses. Table 2 presents the comparison of the nominal income between the treatment group and the comparison group. In 2005, just after the launch of the CRVDP, the average farm income per household in the program implemented areas and non-implemented areas was similar. On the other hand, in 2015, after the termination of the program, farmers in the program implemented areas became relatively richer than the population living in the program not-implemented areas.

**Table 2.** Average farm household income by policy implementation. (Unit: KRW, Thousand).

| | | 2005 | 2015 | Change |
|---|---|---|---|---|
| Nominal Income | | | | |
| | Implemented | 14,648 (USD 13,000) | 22,158 (USD 20,000) | 51.27% |
| | Not-Implemented | 14,521 (USD 13,000) | 19,686 (USD 18,000) | 35.57% |

Source: Authors' calculations based on data from Statistics Korea (2005, 2015) [57].

In terms of a change in farm income before and after the program intervention, the income growth rate of the implemented areas was much higher than that of not-implemented areas. In the period

between 2005 and 2015, the program implemented areas experienced an increase in nominal income by 51.27%, while that of program not-implemented areas was 35.57% during the same period (paired t-test was employed to determine statistically significant differences between 2005 and 2015, and implemented and not-implemented areas. Both results were highly significant at $p < 0.01$). Such results suggest that the CRVDP may have generated additional income for farm households in the program supported areas. In this study, the Heckman selection model and the Blinder–Oaxaca decomposition were used to extrapolate whether the difference in farm income between the implemented areas and non-implemented areas was caused by participation in the government program. The validity of such an analysis can be strengthened by comparing how the difference in outcomes had changed after the CRVDP within the program implemented areas.

*5.2. Cross-Sectional Evaluation on Making Farm Income*

5.2.1. Comparison of Farm Income between the Program Implemented and Not-Implemented Areas

Table 3 reports the results obtained by employing the Heckman selection model to compare outcome estimates of the program implemented areas and not-implemented areas. The first stage binomial probit analysis as presented in Columns (1) and (5) in Table 3 reveals that the effect of the independent variables on the probability of program participation between the two areas was similar in 2005 and 2015 in terms of coefficient signs and statistical significance. The probability of living in the program implemented areas is lower when the householder is older (AGE2→AGE5) as compared to householders who are younger than 34 (AGE1). Generally, the younger the age, the easier it is to acquire and utilize new skills and knowledge. Therefore, this observation is reasonable since younger farmers are more likely to be cognizant of the importance of the program and are more receptive to try new initiatives.

In terms of gender (GENDER), farm households with female householders are more likely to live in program implementation areas than male householders. The number of family members is found to be negatively associated with the probability of living in program implemented areas as shown by the decreasing trend of coefficients as the number of family members increases (HHNUM1 → HHNUM2). Also, a negative association was found between the educational level of the farm householder and the probability of living in the program implemented area. Farm households whose householder has an educational level of below high school (EDU1) or a high school diploma or two-year college degree (EDU2) were more likely to live in the program implemented areas than those with a four-year university or higher degree (EDU3). Such observation may be associated with the fact that sarcastic criticisms on the effectiveness of the CRVDP are more prevalent among people with a higher educational background.

Concerning experience in farming as a career, householders with more than 21 years of experience (EXP3) are more likely to be found in the program implemented areas compared to those with 10 years or less experience (EXP1). However, such evidence was not found for the group with more than 11 years and less than 20 years of experience (EXP2). Lastly, a positive correlation between the possession of farm machinery and residency in the program implemented areas was found. The estimated results of the second stage of the Heckman selection model using the MLE to address the problem of sample selectivity are presented in Columns (2) and (3) for the pre-program period and Columns (6) and (7) for the post-program period. The results present the effect of the independent variables on the probability of making higher farm income, which shows the determinants of farm income in each group. Overall, direction and magnitude of the effects of socio-economic and demographic characteristics such as age, gender, marital status, family size, and career experience on the probability of earning farm income were found to be similar for both groups in the periods before and after the program implementation.

**Table 3.** Estimation results of the Heckman selection model

| | Before Implementation (2005) | | | | After Implementation (2015) | | | |
| --- | --- | --- | --- | --- | --- | --- | --- | --- |
| | (1) | (2) | (3) | (4) | (5) | (6) | (7) | (8) |
| | 1st Stage | 2nd Stage Implemented | 2nd Stage Not-Implemented | *t*-Test | 1st Stage | 2nd Stage Implemented | 2nd Stage Not-Implemented | *t*-Test |
| INTERCEPT | −0.8456 *** | 13.2356 *** | 11.2578 *** | 6.7481 *** | −0.9253 *** | 13.6643 *** | 12.4904 *** | 3.1718 *** |
| AGE2 | −0.0907 ** | | | | −0.0073 | | | |
| AGE3 | −0.1844 *** | | | | −0.1220 | | | |
| AGE4 | −0.2445 *** | | | | −0.1875 ** | | | |
| AGE5 | −0.2423 *** | | | | −0.2236 *** | | | |
| AGE | | 0.0888 *** | 0.0679 *** | 2.2120 ** | | 0.0720 *** | 0.0366 *** | 3.1701 *** |
| AGE_SQ | | −0.0009 *** | −0.0007 *** | −2.4212 ** | | −0.0007 *** | −0.0004 *** | −3.5679 *** |
| GENDER | −0.0930 *** | 0.3653 *** | 0.3826 *** | −0.3904 | −0.0508 *** | 0.3304 *** | 0.3125 *** | 0.4079 |
| MARRY | | 0.2806 *** | 0.2909 *** | −0.2757 | | 0.1562 *** | 0.1635 *** | −0.1945 |
| HHNUM1 | 0.2224 *** | | | | 0.1920 *** | | | |
| HHNUM2 | 0.0884 *** | | | | 0.0609 ** | | | |
| HHNUM | | 0.1091 *** | 0.0556 *** | 5.3679 *** | | 0.1228 *** | 0.0489 *** | 5.7839 *** |
| EDU1 | 0.2976 *** | | | | 0.2213 *** | | | |
| EDU2 | 0.1825 *** | | | | 0.1150 *** | | | |
| EDUY | | −0.0017 | 0.0465 *** | −6.7113 *** | | 0.0062 | 0.0445 *** | −4.4015 *** |
| EDUY_SQ | | 0.0029 *** | −0.0021 *** | 9.5423 *** | | 0.0010 ** | −0.0028 *** | 6.8621 *** |
| EXP2 | −0.0042 | | | | 0.0300 | | | |
| EXP3 | 0.0534 *** | | | | 0.1619 *** | | | |
| NEW | | −0.6321 *** | −0.6339 *** | 0.0358 | | −0.4565 *** | −0.4913 *** | 0.7074 |
| MECH | 0.2958 *** | | | | 0.3086 *** | | | |
| INFO | | 0.5306 *** | 0.5696 *** | −1.2029 | | 0.3733 *** | 0.3391 *** | 1.1436 |
| OTHER | | 0.1009 *** | 0.1941 *** | −2.7115 *** | | 0.5241 *** | 0.6066 *** | −2.7609 *** |
| CROP1 | | 0.2821 *** | 0.4278 *** | −4.2341 *** | | 0.1992 *** | 0.2312 *** | −0.7997 |
| CROP2 | | 0.8897 *** | 0.9685 *** | −1.8327 * | | 0.7904 *** | 0.6823 *** | 2.4538 ** |
| CROP3 | | 0.6886 *** | 0.7760 *** | −2.3379 ** | | 0.4378 *** | 0.5075 *** | −1.7303 * |
| CROP5 | | 1.2110 *** | 1.5066 *** | −6.0624 *** | | 1.6291 *** | 1.7152 *** | −1.4403 |
| S_PLACE1 | | 1.1357 *** | 1.4943 *** | −8.9165 *** | | 1.3646 *** | 1.4971 *** | −3.2972 *** |
| S_PLACE2 | | 0.9768 *** | 1.2178 *** | −7.8822 *** | | 1.1725 *** | 1.2903 *** | −3.8108 *** |
| S_PLACE3 | | 0.9457 *** | 1.1145 *** | −5.6843 *** | | 1.1435 *** | 1.2361 *** | −2.6468 *** |
| S_PLACE5 | | 1.1185 *** | 1.1099 *** | 0.2180 | | 0.4369 *** | 0.4352 *** | 0.0372 |
| SIGMA | | 1.8546 *** | 1.3316 *** | | | 1.6411 *** | 1.4430 *** | |
| RHO | | −0.9083 *** | −0.6018 *** | | | −0.8244 *** | −0.7693 *** | |
| −2LL | | 153,810 | 235,722 | | | 124,908 | 195,886 | |
| AIC | | 153,878 | 235,790 | | | 124,976 | 195,954 | |
| N | 69,013 | 21,951 | 47,062 | | 57,030 | 17,686 | 39,344 | |

Note: * $p < 0.1$, ** $p < 0.05$, *** $p < 0.01$.

In regard to age of householders (AGE), the probability of farm income increases as the householder's age increases, regardless of the residential areas. The older the householder, the higher the householder's chance of earning farm income; but since the squared term of age is negative, the marginal effect is expected to be a decreasing trend. The non-linear term is included since the effect of age on agricultural productivity that affects farm income could be positive up until a certain threshold age and then negative thereafter [58]. Concerning gender (GENDER), households headed by males were found to have a higher probability to earn farm income than female-headed households.

The number of family members (HHNUM) has a strong, positive effect on farm income. Households of bigger family size are more likely to earn higher farm income than households with fewer members. A positive association between years of education (EDUY) and farm income was apparent only in the program implemented areas. The squared term of education is added as the effects of the formal schooling on agricultural performance differ across countries or contexts [59]. In the case of Korea, the coefficient of its squared term (EDUY_SQ) presents a negative sign through which we can assume that the effect of educational background is non-linear.

Farming experience (NEW) had a positive effect on farm income. The likelihood of earning farm income was found to be lower for householders with less than six years of agricultural experience than that of skilled agriculturalists. On the other hand, the utilization of computer (INFO) and participation in other agriculture-related businesses (OTHER) were found to be positively associated with the chance of farm income earning.

In terms of major crops, the probability of obtaining farm income is higher for farmers who cultivate rice (CROP1), fruits (CROP2), other crops (CROP3), and livestock (CROP5) in comparison to those who are primarily engaged in the cultivation of upland crops (CROP4). Such results are reasonable in the context of Korea where upland crops such as barely, beans, and corns are cultivated in a small-scale by aged farmers. Rice, a staple food in Korea, is the most preferred crop type because of the high level of mechanization and the direct payment policy, but the income of rice farmers is as low as upland crops. On the other hand, livestock farmers earn the highest income from farm activities.

For sales place, farm households that trade in agricultural products through wholesale markets (S_PLACE1), agricultural cooperatives and corporations (S_PLACE2), government agencies and other mediators (S_PLACE3), and retailers and processing companies (S_PLACE5) have a higher earning potential than households selling directly to consumers (S_PLACE4). Such propensity is particularly prominent where the program was not carried out.

The results of the asymptotic *t*-test ($t = \frac{\beta_1 - \beta_2}{\sqrt{se(\beta_2)^2 + se(\beta_2)^2 - 2COV(\beta_1, \beta_2)}}$) that compare the significant differences between the implemented areas and not-implemented areas show whether or not the two groups were profoundly different with respect to each independent variable. Looking at the comparison between the program implemented areas and not-implemented areas in the pre-program period as presented in Column (4) in Table 3, we can extrapolate whether there was an initial difference between the two areas. A statistical difference between the two areas is evident in regard to age, number of household members, years of education, participation in other agriculture-related businesses, all crop types, and sales place—except for those who sell to retailers and processing companies. The result of t-statistics in Column (9) in Table 3 highlights the differences between the program implemented areas and not-implemented areas in the post-program period. It shows that age, age_sq, number of family members, years of education, participation in other agricultural-related businesses, cultivation of fruits and other crops, and sales place (excluding those who trade through retailers and processing companies) affected the chance of earning farm income differently for the program supported areas and unsupported areas.

In the bottom of Table 3, sigma ($\sigma$) and rho ($\rho$) are obtained for each time period. In both time periods, estimates of sigma and rho are statistically significant at 1% level. This implies that a selection bias would have occurred if a simple ordinary linear regression (OLS) model was applied instead of the Heckman selection model, thereby supporting the validity of employing this particular methodology for this study. Moreover, the negative sign of rho ($\rho$) with a strong statistical significance in both periods

highlights a differential gap between the two areas with respect to earning farm income. This implies that the expected farm income of the program not-implemented areas would have been higher if their initial conditions were the same as the program implemented areas. It suggests that rural villages with relatively low potentials to generate farm income were selected as the program beneficiaries.

### 5.2.2. Decomposition for Cross-Sectional Program Effectiveness

Making causal claims is central to policy evaluation. In this study, we apply the Blinder–Oaxaca decomposition to evaluate the causal relationship of the CRVDP on the apparent differential gap in farm income across two groups of areas after the completion of the program. The estimates elicited from the Heckman selection model show that even when farm households in the program implemented areas and not-implemented areas were analogous on average, the expected earnings of the supported areas would have been higher than the unsupported areas. The Blinder–Oaxaca decomposition model is a feasible strategy to derive what explains the average differences on outcomes between the two groups.

The results of the Blinder–Oaxaca decomposition for the cross-sectional analysis employed on the selectivity corrected outcomes are reported in Table 4. In the decomposition model, the difference between the treatment group and the comparison group is explained by the total effect. Further, the total effect is composed of the endowment effect, which results from the differences in independent variables, and the residual effect, which is the remaining effect not explained by the differences in independent variables. Therefore, the treatment effect of the CRVDP on farm income is represented by the residual effect.

**Table 4.** Cross-sectional decomposition on probability of making farm income.

|  | Implemented | | Not-Implemented |
| --- | --- | --- | --- |
| Estimated | 17.3437 | | 15.0675 |
| Hypothetical Estimates | | 17.3328 | |
| Difference | | 2.2762 | |
| Endowment Effect | | 0.0108 | |
| Residual Effect | | 2.2653 | |
| Gap (%) explained by | | | |
| Endowment Effect | | 0.48% | |
| Residual Effect | | 99.52% | |

The difference in the income estimates between the two groups converted into a natural logarithm was 2.2762. The endowment effect that arises from the differences in observed characteristics between the two groups was 0.0108 (0.48%), and the residual effect or the treatment effect was 2.2653 (99.52%). This implies that 99.52% of the difference in farm income between the two groups is likely to be caused by the program implementation. From Table 1, we observed that the difference in farm income on average between the program implemented areas and not-implemented areas was about 247.2 million Won. Based on the results of the decomposition, it can be concluded that 99.52% of the difference in average farm household income was caused by the CRVDP.

### 5.3. Longitudinal Evaluation on Making Farm Income

#### 5.3.1. Comparison of Farm Income before and after Program Implementation

Table 5 provides the estimation results of the Heckman selection model applied solely on the program implemented areas to compare outcomes between the pre- and post-program periods. Given the limitation of using the census data from different years rather than the identical sample to compare the coefficients of two different time periods, there remains a possibility that the sample households from the 2015 data is significantly different from those extracted from the 2005 data. Therefore, the Heckman selection model was employed to correct the selection problem in estimating the net impact of the CRVDP from the longitudinal perspective.

**Table 5.** Estimation results of the Heckman selection model on policy implemented areas.

| Variables | (1) 1st Stage | (2) 2nd Stage Before Implementation | (3) 2nd Stage After Implementation | (4) *t*-Test |
|---|---|---|---|---|
| INTERCEPT | −0.7014 *** | 12.5738 *** | 16.1035 *** | −8.9054 *** |
| AGE2 | 0.0923 | | | |
| AGE3 | 0.5116 *** | | | |
| AGE4 | 0.8579 *** | | | |
| AGE5 | 1.0547 *** | | | |
| AGE | | 0.0511 *** | −0.0070 | 4.7898 *** |
| AGE_SQ | | −0.0007 *** | −0.0002 *** | −4.9902 *** |
| GENDER | −0.1441 *** | 0.3532 *** | 0.3381 *** | 0.3053 |
| MARRY | | 0.3648 *** | 0.2380 *** | 2.8728 *** |
| HHNUM1 | 0.4380 *** | | | |
| HHNUM2 | 0.2256 *** | | | |
| HHNUM | | 0.0764 *** | 0.1337 *** | −4.3269 *** |
| EDU1 | −1.1129 *** | | | |
| EDU2 | −0.4342 *** | | | |
| EDUY | | 0.0581 *** | 0.0538 *** | 0.4544 |
| EDUY_SQ | | −0.0049 *** | −0.0061 *** | 1.7677 * |
| EXP2 | 0.2160 | | | |
| EXP3 | 0.1327 *** | | | |
| NEW | | −0.6973 *** | −0.4704 *** | −3.7039 * |
| MECH | 0.3416 *** | | | |
| INFO | | 0.5598 *** | 0.3629 *** | 5.1936 *** |
| OTHER | | 0.0581 ** | 0.4841 *** | −10.9759 *** |
| CROP1 | | 0.3526 *** | 0.1575 *** | 4.5198 *** |
| CROP2 | | 1.0523 *** | 0.7192 *** | 6.5673 *** |
| CROP3 | | 0.7495 *** | 0.3905 *** | 8.0011 *** |
| CROP5 | | 1.3135 *** | 1.4506 *** | −2.1904 ** |
| S_PLACE1 | | 1.2105 *** | 1.3476 *** | −2.8228 ** |
| S_PLACE2 | | 1.0813 *** | 1.1741 *** | −2.5230 ** |
| S_PLACE3 | | 1.0537 *** | 1.1328 *** | −2.0319 ** |
| S_PLACE5 | | 1.1643 *** | 0.3813 *** | 15.3646 *** |
| SIGMA | | 1.2845 *** | 1.4427 *** | |
| RHO | | −0.5347 *** | −0.7213 *** | |
| −2LL | | 120,566 | 105,740 | |
| AIC | | 120,634 | 105,809 | |
| N | 39,679 | 22,114 | 17,376 | |

Note: * $p < 0.1$, ** $p < 0.05$, *** $p < 0.01$.

The first stage binomial probit analysis shown in Column (1) in Table 5 presents the effect of each independent variable on the probability of being in the sample of the post-program period. The positive signs of age variables show that householders in the post-program period are more likely to belong to the older group as seen by the incrementing coefficients for older age groups (AGE2 → AGE5). This implies that over time, from 2005 to 2015, the age of householders in the program-supported areas had increased, which seems to be reasonable in light of the acceleration of aging in rural areas.

The negative coefficient of the gender variable (GENDER) indicates that farm households headed by male householders are more likely to be households of the post-program period. This means that the number of households with male householders had increased over the course of the program within the program implemented areas. With respect to householder's level of education, the households in the post-program period are less likely to have lower levels of education since the lower the educational level, the bigger the negative coefficients of the estimation. From this result, we can infer that more educated farmers participated in the CRVDP over time.

Concerning experience in farming, there were more experienced farmers after the program in the areas that received the government support. A positive coefficient for those with more than 20 years of experience (EXP3) proves this observation, whereas statistical significance was not obtained for those with more than 10 or less than 20 years of experience (EXP2). Lastly, households in the post-program period in the program implemented areas were more prone to possess agricultural machinery (MECH) than households in the pre-program period in the same areas.

The second stage estimation of the sample selection regression results are reported in Columns (2) and (3) in Table 5, which highlights the effect of determinants of farm income in the pre- and post-program periods. After adjusting for selection, coefficients of all variables except for age (AGE) show that the statistical significance as well as the direction of impact on farm income are consistent before and after the program.

In the period before the program implementation, the age variable (AGE) shows a positive sign, while the coefficient of its squared term (AGE_SQ) presents a negative sign. It indicates that before the program, the probability of earning farm income was higher for households with older householders, although the effect lessoned as people get older. However, in the period after the program, statistical significance was not found for the age variable (AGE) while age squared (AGE_SQ) had a significantly negative effect on farm income. This suggests that unlike the households in the pre-program period, the age of farming householders did not have a non-linear effect on farm income in the post-program period.

In both before- and after-program implementation, households whose householder is male (GENDER) and married (MARRY) and with a bigger family size (HHNUM) had a higher chance to earn farm income in the program implemented areas. The impact of marital status on farm income was stronger in the pre-program period, while that of family size was greater in the post-program period. In respect to years of education (EDUY) and its squared term (EDUY_SQ), the result indicates that the probability of earning farm income increased as the years of education increased, but this effect lessened with the lapse of time.

Households who are new to farming (NEW) had a lower chance of earning farm income, and such negative effect is found to be slightly weaker after the implementation of CRVDP. On the contrary, households with a higher frequency of using computer (INFO) had a greater chance to earn farm income, the effect of which became relatively weaker after the program implementation. We can reasonably assume that such trend is observed with the penetration of computer into general households over time since obtaining information became easier even for novice farmers and the comparative advantage of utilizing computer weakened. Also, households who are involved in other agriculture-related businesses (OTHER) were found to be more likely to earn farm income, the effect of which was shown to be stronger after the implementation of the program.

In regard to major crop type, households that grow rice (CROP1), fruits (CROP2), other crops (CROP3), and livestock (CROP5) show a higher probability to make more farm income than those that grow upland crops (CROP4). Moreover, farm households that trade agricultural products through wholesale markets (S_PLACE1), agricultural cooperatives and corporations (S_PLACE2), government agencies and other mediators (S_PLACE3), and retailers and processing companies (S_PLACE5) were found to have a higher probability to earn income than households selling directly to consumers (S_PLACE4). The difference in the influence of these variables on farm income before and after the program implementation was particularly stronger in the case of direct sales (S_PLACE5).

The result of the asymptotic t-test in Column (4) in Table 5 presents the significance of differences in outcomes between pre- and post-program periods. The two years were different with respect to age, age squared, marital status, number of family members, squared term of education years, utilization of computer, participation in other agriculture-related businesses, crop types, and sales places. The estimates of sigma ($\sigma$) and rho ($\rho$) presented in the bottom of Table 5 are found to be statistically significant at the 1% level. This implies that sample selection bias could have been problematic in the before-and-after comparison if the correction was not addressed by the Heckman

selection model. As in the case of the cross-sectional analysis, estimates of the correction factor calculated by the product of sigma ($\sigma$) and rho ($\rho$) is negative. This indicates that in the program implemented areas, the expected farm income of the farm households prior to the program participation would have been higher if the same households were exposed to the same contextual setting as in the post-program period.

### 5.3.2. Decomposition for Longitudinal Program Effectiveness

Table 6 shows the estimation of outcome gap in pre- and post-program periods using the Blinder–Oaxaca decomposition approach after adjusting for selection. The result finds that the program supported areas have benefited from the CRVDP when their chance of earning farm income before and after the program implementation is compared.

**Table 6.** Longitudinal decomposition on probability of making farm income.

|  | After Implementation | Before Implementation |
|---|:---:|:---:|
| Estimated | 16.7145 | 15.3670 |
| Hypothetical Estimates | 16.9207 | |
| Difference | 1.3475 | |
| Endowment Effect | −0.2110 | |
| Residual Effect | 1.5585 | |
| Gap (%) explained by | | |
| Endowment Effect | −15.66% | |
| Residual Effect | 115.66% | |

The total difference in the estimated values between before- and after-program implementation is 1.3475, of which 115.66% is explained by the residual effect. The negative endowment effect (−15.66%) implies that the probability of earning farm income has become less favorable over time. This can be explained by the current rural issues such as aging and out-migration of the youth population. This signifies that if the program was not carried out in the program implemented areas, the chance to make farm income for households in these areas would have reduced due to the changes in characteristics of farm households over the course of the program period. However, fortunately, farm households were able to sustain the level of farm income thanks to the execution of the CRVDP. The negative endowment effect has been offset by the residual effect, and, therefore, the program had positive influence on rural households' income even under deteriorated conditions.

## 6. Discussion and Conclusions

The decline of rural areas is causing the problems of the rural exodus and aging, which, in turn, further exacerbates rural economic crises at an unprecedented rate. Such problems in rural areas are rapidly showing up on government policy agendas around the world. In Korea, the vast sum of investment allocated to the agricultural and rural sector since 2004 is an effort by the government to revive rural communities for sustainable development. Despite the consensus on the necessity of rural revitalization, economic efficiency or value for money has been called into question both at home and abroad. In particular, a skeptical view has been heightened for the case of the Comprehensive Rural Village Development Program in Korea. This is related to the tendency of policymakers to give lower priority to subjective assessment of policy outcomes once the policy intervention is terminated. However, an accurate and valid assessment of a precedent policy is fundamental in designing future policies and programs to ensure their quality and effectiveness.

In the light of this background, this study conducted an ex-post evaluation of a net impact of the CRVDP on a targeted outcome. Rural households' farm income was chosen as the study's ex-post quantifiable indicator of program effect, and the impact of the program on this indicator was analyzed. Farm income is chosen not only because it is one of the explicit goals of the program, but also because

attaining higher farm income is important in a broader rural context; the low profitability from farming is one of the significant factors that force the youth to abandon agriculture and seek for economic opportunities in urban areas. Some of the negative consequences of such trend are the degradation of residential environments with vacant houses and abandoned farmland, limited provision of public services that further deteriorates standard of living, and environmental problems that arise from inadequate management of agricultural wastes, all of which, in turn, undermine the sustainability of rural communities. Moreover, attracting young people to farming by a means of higher farm income can be crucial for rural sustainability in Korea where eco-friendly agriculture is dominantly led by the young farmers.

The results of our analysis from both the cross-sectional and longitudinal perspectives support that the CRVDP had a positive impact on farm income of rural households. Such positive results in terms of farm income highlight the importance of the community-initiated, integrated rural development approach. The injection of vast amounts of money into rural areas that are top-down in essence with fragmented goals can be counterproductive as shown by rural revitalization strategy that had resulted in numerous vanity projects in the case of Japan [60], and also the controversial Millennium Villages Project, which has been widely criticized as failed fantasy [61,62]. In contrast, the CRVDP had ensured that local actors collectively diagnose their own problems and come up with appropriate solutions; also, communities had to submit their proposals and those proposals were selected on a competitive base, which strengthened ownership at community level. Thus, this type of rural development program that takes bottom-up, integrated multi-sectoral approach should be more widely promoted. Such programs can be enhanced if they are supplemented with programs that foster capacities of rural residents to keenly identify the current problems and come up with more creative ideas. Also, considerations for the economic, social, and environmental sustainability need to be enhanced at the proposal stage to ensure the long-term viability of the program. The positive findings also suggest that the CRVDP can be referred to as an exemplary case for countries and international organizations interested in developing rural development programs on a large-scale.

Unlike other quasi-experimental evaluation tools that are often unsuitable due to the coverage of available databases, we were able to attain robust results using secondary data by employing the combined use of the Heckman selection model and the Blinder–Oaxaca decomposition. The findings can be summarized as follows.

First, the study found that in both with-and-without and before-and-after analyses of program impact, drawing a valid causal inference would have been difficult in the absence of the attempt to address potential selection bias. The negative sign of the correction factor was found in both cross-sectional and longitudinal comparisons. An application of general linear analysis on farm income neglecting unobserved households' characteristics and time lag would have caused statistical errors. In this study, such possible errors were addressed by employing the Heckman selection model.

Second, in a cross-sectional analysis that compared the program implemented areas with the not-implemented areas, the probability of earning farm income is much higher for the program implemented group than for the program not-implemented group. By decomposing the total difference (2.2762) in the propensity of making higher farm income between the two groups, we found that only 0.48% is attributable to endowment effect, while 99.52% is explained by residual difference across the two groups. This finding implies that endowed resources or observed characteristics controlled in the model do not explain the existing differences between the two groups. On the other hand, the residual effect that represents the average treatment effect of the program proves a strong positive net effect on earning farm income in the program implemented areas.

Third, in a longitudinal analysis exploring outcome differences between pre- and post-program periods, a higher propensity of making farm income is found after the termination of the program. Decomposing the total difference (1.3475) in the probability of making higher farm income, we found that −15.66% is attributable to endowment effect and 115.66% is explained by the residual difference between the post- and the pre-program samples. A negative endowment effect implies that the probability

of earning farm income became less favorable after the program's implementation. Nevertheless, the negative endowment effect is offset by a strong residual effect (115.6%). Such results signify that the program was carried out in relatively poor areas with low potential and had a positive effect on making farm income in those areas.

Finally, based on the findings above, this study concludes that the CRVDP generated a significant positive impact on enhancing households' chance to make farm income in rural areas. The study also finds that the selection of beneficiaries was appropriate. If there had been no government support, the program implemented areas could have been exposed to a difficult situation in terms of income earning.

The significance of this study lies in its attempt to introduce an alternative, flexible quantitative evaluation tool for using readily available data from ex-post perspective. This study analyzed the effectiveness of a rural program by applying a quasi-experimental design that has been rarely utilized in assessing agricultural and rural policies. In particular, unlike other quasi-experimental evaluation tools that are often unsuitable due to the coverage of available databases, we were able to attain robust results using secondary data by employing the combined use of the Heckman selection model and the Blinder–Oaxaca decomposition. In this respect, this study is expected to become a useful reference in conducting ex-post evaluations for not only the agricultural and rural sector, but for other public policies in various sectors as well.

This study can contribute to the policy evaluation literature in search of more flexible ways to rigorously estimate causal effects from observational data. However, the study has limitations as follows. First of all, the CRVDP was implemented at the village level, which renders it necessary to control regional characteristics at the village level; however, we were not able to reflect such variables in the model due to limited data. In addition, this study narrowed down the target indicator of program effectiveness to farm income. However, since the CRVDP encompasses other goals, such as the quality of life and non-farm income, an empirical study thoroughly investigating the overall effects by constructing an index covering all other main goals of the program would allow the estimation of broader program impacts.

**Author Contributions:** Conceptualization, writing—review and editing, E.C. and S.L.; methodology, E.C.; formal analysis, J.P.; software, J.P.; data curation, E.C.; validation, E.C., J.P. and S.L.; writing—original draft preparation, E.C.; supervision, S.L. All authors have read and agreed to the published version of the manuscript.

**Funding:** This research received no external funding.

**Conflicts of Interest:** The authors declare no conflict of interest.

## Appendix A. The Heckman Selection Model

The estimation process of the sample selection model involves two stages. In the first stage, the probability of selection into the sample is estimated by a probit regression. These probabilities are observed indirectly by a binary variable that indicates whether the case is included in the sample (=1) or not (=0). In this study, we want to estimate the outcome (farm income) Equation (A1). An auxiliary model for the process of generating latent $z^*$ that describes the propensity to participate in the program is Equation (A2). The observed counterpart of $z^*$ expressed as z is determined by Equation (A3). Values of y and x are only observed when z equals 1.

$$\text{Outcome equation}: \ y = \beta' x + \varepsilon \tag{A1}$$

$$\text{Selection equation}: \ z^* = \alpha' w + u \tag{A2}$$

$$z = 1 \text{ if } z^* > 0 \text{ and } z = 0 \text{ if } z^* \leq 0 \tag{A3}$$

Given the non-randomness of program participation, there is a high likelihood that $\varepsilon$ and $u$ are correlated. Heckman [42] proposes the likelihood estimation method by way of a two-step method. For the subsample with a positive y, the conditional expectation of $y$ is given by Equation (A4):

$$E[y_i|x_i,\ in\ sample] = E[y_i|x_i,\ Z = 1]$$

$$= E[y_i|x_i,\ \alpha'w_i + u_i > 0]$$

$$= \beta'x_i + E[\epsilon'|u_i > -\alpha'w_i] \tag{A4}$$

$$= \beta'x_i + (\rho\sigma_\epsilon\sigma_u)\left\{\frac{\phi(-\alpha'w_i)}{1 - \Phi(-\alpha'w_i)}\right\} \tag{A5}$$

$$= \beta'x_i + (\rho\sigma_\epsilon\sigma_u)\left\{\frac{\phi(-\alpha'w_i)}{\Phi(\alpha'w_i)}\right\} \tag{A6}$$

Assuming a bivariate normal distribution of $\varepsilon$ and $u$, the conditional expectation of the error term can be expressed as Equations (A5) and (A6) where $\phi$ is standard normal probability density function and $\Phi$ is cumulative standard normal distribution function.

## Appendix B. The Blinder–Oaxaca Decomposition Technique

The detailed mathematical expression of the Blinder–Oaxaca decomposition technique is as follows:

$$E(Y_A) - E(Y_B) = \sum_{j=1}^{k}\beta_j^A\overline{X}_j^A - \sum_{j=1}^{k}\beta_j^B\overline{X}_j^B$$

$$= \sum_{j=1}^{k}\beta_j^A\left(\overline{X}_j^A - \overline{X}_j^B\right) + \sum_{j=1}^{k}\beta_j^A\overline{X}_j^B - \sum_{j=1}^{k}\beta_j^B\overline{X}_j^B \tag{A7}$$

$$= \sum_{j=1}^{k}\beta_j^A\left(\overline{X}_j^A - \overline{X}_j^B\right) + \sum_{j=1}^{k}\overline{X}_j^B\left(\beta_j^A - \beta_j^B\right) \tag{A8}$$

The left-hand side of Equation (A7) denotes the difference in the program impact between program implemented areas $(Y_A)$ and not-implemented areas $(Y_B)$. Equation (A8) is obtained by the mathematical decomposition of Equation (A8), which distinguishes the differences in policy evaluation estimates between the two groups.

## Appendix C. Descriptive Statistics of Sample Farm Households

**Table A1.** Descriptive statistics of implemented and not-implemented areas in 2005 and 2015.

| | 2005 | | | | 2015 | | | |
| --- | --- | --- | --- | --- | --- | --- | --- | --- |
| | Not-Implemented | | Implemented | | Not-Implemented | | Implemented | |
| | Mean | S.D | Mean | S.D | Mean | S.D | Mean | S.D |
| FARM INCOME | 15.5004 | 1.5118 | 15.5567 | 1.4909 | 15.6327 | 1.5225 | 15.8342 | 1.4908 |
| AGE | 60.8388 | 11.0930 | 61.4844 | 11.0334 | 64.9052 | 10.9964 | 65.9798 | 10.9077 |
| AGE_SQ | 3824.41 | 1315.25 | 3902.06 | 1313.56 | 4333.60 | 1398.09 | 4472.30 | 1402.48 |
| GENDER | 0.8380 | 0.3684 | 0.8270 | 0.3782 | 0.8314 | 0.3744 | 0.8202 | 0.3840 |
| MARRY | 0.8021 | 0.3984 | 0.7883 | 0.4086 | 0.7887 | 0.4083 | 0.7665 | 0.4231 |
| HHNUM | 7.2310 | 4.2797 | 6.5789 | 4.1412 | 8.5965 | 4.2956 | 7.7809 | 4.1896 |
| EDUY | 70.6038 | 63.7423 | 60.4305 | 56.8310 | 92.3516 | 71.1499 | 78.0942 | 64.3175 |
| EDUY_SQ | 2.7299 | 1.3952 | 2.5427 | 1.3052 | 2.3779 | 1.1620 | 2.2221 | 1.0922 |
| NEW | 0.0538 | 0.2256 | 0.0395 | 0.1947 | 0.0644 | 0.2454 | 0.0560 | 0.2300 |
| INFO | 0.1118 | 0.3151 | 0.1128 | 0.3163 | 0.2014 | 0.4011 | 0.1877 | 0.3905 |
| OTHER | 0.0866 | 0.2813 | 0.0864 | 0.2809 | 0.1852 | 0.3885 | 0.1762 | 0.3810 |
| CROP2 | 0.5125 | 0.4998 | 0.4814 | 0.4997 | 0.4252 | 0.4944 | 0.3889 | 0.4875 |
| CROP3 | 0.1211 | 0.3263 | 0.1164 | 0.3207 | 0.1755 | 0.3804 | 0.1718 | 0.3772 |
| CROP4 | 0.2124 | 0.4090 | 0.2322 | 0.4222 | 0.2572 | 0.4371 | 0.2859 | 0.4518 |
| CROP5 | 0.0696 | 0.2545 | 0.0678 | 0.2515 | 0.0510 | 0.2199 | 0.0574 | 0.2326 |
| S_PLACE1 | 0.1058 | 0.3076 | 0.0991 | 0.2988 | 0.1091 | 0.3117 | 0.1197 | 0.3246 |
| S_PLACE2 | 0.2617 | 0.4396 | 0.3065 | 0.4611 | 0.3602 | 0.4801 | 0.3847 | 0.4865 |
| S_PLACE3 | 0.3184 | 0.4659 | 0.3474 | 0.4762 | 0.1579 | 0.3646 | 0.1902 | 0.3925 |
| S_PLACE5 | 0.1103 | 0.3132 | 0.0924 | 0.2896 | 0.0967 | 0.2955 | 0.0852 | 0.2791 |
| N | 47,062 | | 21,951 | | 39,344 | | 17,686 | |

**Table A2.** Descriptive statistics for implemented areas in 2005 and 2015.

| | 2005 | | 2015 | |
| --- | --- | --- | --- | --- |
| | Mean | S.D | Mean | S.D |
| Farm Income | 15.8159 | 1.4812 | 15.8448 | 1.4875 |
| AGE | 61.5581 | 10.9929 | 65.8776 | 10.9604 |
| AGE_SQ | 3910.23 | 1308.42 | 4459.99 | 1406.17 |
| GENDER | 0.8261 | 0.3790 | 0.8177 | 0.3861 |
| MARRY | 0.7864 | 0.4099 | 0.7646 | 0.4243 |
| HHNUM | 6.5969 | 4.1304 | 7.7987 | 4.1864 |
| EDUY | 60.5789 | 56.9942 | 78.3436 | 64.2047 |
| EDUY_SQ | 2.5484 | 1.3222 | 2.2204 | 1.0925 |
| NEW | 0.0382 | 0.1916 | 0.0520 | 0.2220 |
| INFO | 0.1094 | 0.3121 | 0.1883 | 0.3910 |
| OTHER | 0.0861 | 0.2805 | 0.1774 | 0.3820 |
| CROP2 | 0.4761 | 0.4994 | 0.3844 | 0.4865 |
| CROP3 | 0.1179 | 0.3225 | 0.1727 | 0.3780 |
| CROP4 | 0.2322 | 0.4222 | 0.2812 | 0.4496 |
| CROP5 | 0.0696 | 0.2545 | 0.0619 | 0.2410 |
| S_PLACE1 | 0.0989 | 0.2985 | 0.1138 | 0.3176 |
| S_PLACE2 | 0.3067 | 0.4611 | 0.3913 | 0.4881 |
| S_PLACE3 | 0.3445 | 0.4752 | 0.1859 | 0.3890 |
| S_PLACE5 | 0.0931 | 0.2905 | 0.0885 | 0.2840 |
| N | 22,114 | | 17,376 | |

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
