# Peer review of "The Effect of the Comprehensive Rural Village Development Program on Farm Income in South Korea"

_sustainability, doi:10.3390/su12176877_

Round 1
Reviewer 1 Report
This is a innovative methodological approach to the problem of post-policy and post-intervention evaluation, something sorely lacking in practice and academia, as you point out. The work contributes to this gap. My main suggestion (hence my 'can be improved' mark above) is to clearly link, conceptually, an increase in rural farm incomes to sustainability. There is literature no doubt that would help here, but also the context of S. Korea and rural livelihoods. Does an increase in rural farm incomes prevent the flight to the cities? Does it maintain the land in a more sustainable way? How does it affect these livelihoods beyond income generation? This is mainly a reminder that neoliberal development theory that focuses only on income and economic growth misses other elements of development and equity, and in particular, the environment.
Reviewer 2 Report
Between 2004 and 2013 the Comprehensive Rural Village Development Program (CRVDP) was implemented (amounting to 119 trillion Korean Won (equivalent to 108 Billion USD); with goals to improve the residential environment and the income of rural residents. Data availability difficulties and the fact that ex-post policy evaluation is “forgotten phase of planning” usually contribute to limited studies on the ex-post effects of such policy interventions.
The authors select the CRVDP, as a case study and conducted an ex-post evaluation to quantify the effects of the program in boosting the farm income of the rural population. In order to do so, the study constructs econometrics models incorporating the Hackman selection model and the Blinder-Oaxaca decomposition method.
The paper correctly considers a quasi-experimental strategy. When there is a policy intervention/treatment and in the absence of a natural experiment in which control and treated units are assigned on a total random basis, we need to have a strategy that clarifies the assignment of the treated units to the treatment. The authors use a Heckman selection model which sheds light upon the characteristics of the observations and studies the selection bias, meaning if a specific set of characteristics leads to a higher propensity to get the treatment (which will bias out estimates of the effects). The second stage of the provided analysis is a Blinder-Oaxaca decomposition in which the total estimated effect is divided by two: the residual effect is the effect reflected by factors other than the differences in the independent variables between the two groups in the same time period thus, the residual effect captures the average treatment effect.
Below, please find my comments which I believe will improve the paper substantially:
The paper will improve if the authors provide some reference to the fundamental/well-known papers of the quasi-experimental literature (diff-in-diff, matching (synthetic control method), propensity score…).
The setup of the model for the cross-sectional analysis is well-explained in Sections 3.1 and 3.2, yet not comprehensively explained for not for the longitude study. The models and methods are well-known, and I recommend making these sections much shorter and moving the details of these sections to the appendix. The findings of the paper are more interesting, the interested audience could explore the methodology in the appendix.
Why in the selection of the sample from the Census data, the choice is a selection of 10% in the implemented are and 5% in the not implemented are? What is the justification of this choice? Please clarify in the paper.
From Table 3, the younger the householder, the higher the probability of living in the implemented area compared to householders of 65 or higher age. However meanwhile, the experience variables coefficients state that those farmers with higher than 21 years of experience are more present in the implemented areas compared to less experienced. What I recommend is to leave the same dummy category out (to avoid collinearity) AGE1 and EXP1 and not AGE5 and EXP1.
On page 13, the marginal effect of the variable age-squared is too small yet significant. I leave it to the authors to decide but the variable can be removed from the model.
In the introduction and the abstract, paper will be improved if you briefly highlight the content of Section 5.2.1. in which you study the dynamics of the socio-economic characteristics of the householders (and type of the agricultural activity) and how they cause different effectiveness of the policy in the implemented versus not-implemented areas.
On Table 3 match the style on the first few rows. You are missing a horizontal line. Also, on Table 5.
On line 224 there is a typo z=0 if z*<=0
Reviewer 3 Report
I think this is a nice piece of work.
Some concerns:
- Justification for the use of Heckman and Blinder-Oaxaca can be made clearer with review of previous attempts adopting the methods and their limitations.
- I wonder of similar level of CRVDP program outside Korea that can be discussed for the benefit of global sustainable development. I guess it is expected from the program of the positive results, but what can we learn from it? develop another program? transferability of the program? UN SD agenda?
- I also find that the focus is very much on comparing socio-economic and demographic differences rather than agricultural/crops types which I think would be of interest to demonstrate the uniqueness of Korean agriculture landscape.
- the naming of the variables as presented in the article can be made clearer with using their real name rather than the code used in the statistical software. Btw, what statistical software do you use?
- there are a few typos, such as Hackman, heckit, etc. please check.
Reviewer 4 Report
This is an interesting study on rural development program evaluation, with significant methodological contribution. However, the results are barely discussed. The study also lacks compelling policy implication. This is especially important because of the policy relevance of the topic. Following are detailed comments.
Comments
- Mention should be made to rival methods that have been used specifically in evaluating rural development projects. AHP, in particular, has been applied extensively to evaluate rural development programs in many contexts, (see e.g. rural livelihoods in Ghana (Baffoe 2019). What makes Hackman selection model and the Blinder-Oaxaca decomposition method superior over existing approaches? A robust explanation and justification is needed.
- Equation 1c is confusing. It says z=1 if ?∗>0 and z=1 if ?∗≤0. What is the difference? In Probit modelling, x is usually = 1 if x is observable, and x = 0, if otherwise. Greater than 0 is not the same as less than or equals to 0. I sense an error here. Meanwhile, all the variables in the three equations need to be explained, so that ordinary readers can understand what they mean.
- Explanations and justifications on the independent variables are lacking. The authors claim that “the probable determinants that affect the likelihood of earning farm income based on the previous literature ….”. This is not enough to justify why the variables were selected. There should be a review section on their appropriateness – thus, a robust theoretical underpinning is needed. And why age square and education year square were added to the model? This should be explained for readers who not familiar with statistical modelling to understand.
- How significant is the differences in income changes in Table 2? A statistical test may be helpful here.
- The results are interesting but they are barely discussed. It needs to be discussed by placing them in the broader rural development literature. The current form is too shallow and devoid of critical interrogation.
- The methodological significance of the study is welcome, however, a proper policy implication is needed, given the policy relevance of the topic. What can policy and countries with similar developmental characteristics learn from the analysis?
- The study would also benefit from study limitations and recommendation for future work/s. The former even becomes more imperative given that the two datasets were not comparable. Some thoughts on the data challenges and variable limitations would be welcomed.
Reference
Baffoe, G. (2018). Exploring the utility of Analytic Hierarchy Process (AHP) in ranking livelihood activities for effective and sustainable rural development interventions in developing countries. Evaluation and Program Planning, 72, 197-204.
Round 2
Reviewer 4 Report
I am satisfied with the responses provided by the authors.